# Development of an epigenetic tetracycline sensor system based on DNA methylation

**Timo Ullrich, Sara Weirich, Albert Jeltsch** 📷 *

Department of Biochemistry, Institute of Biochemistry and Technical Biochemistry, University of Stuttgart, Stuttgart, Germany

* albert.jeltsch@ibtb.uni-stuttgart.de

**Data Availability Statement:** All relevant data are within the paper and its Supporting Information files.

**Funding:** This work has been supported by the Deutsche Forschungsgemeinschaft (DFG) grant JE

## Abstract

Bacterial live cell sensors are potentially powerful tools for the detection of environmental toxins. In this work, we have established and validated a flow cytometry readout for an existing bacterial arabinose sensor system with DNA methylation based memory function (Maier et al., 2017, Nat. Comm., 8:15336). Flow cytometry readout is convenient and enables a multiparameter analysis providing information about single-cell variability, which is beneficial for further development of sensor systems of this type in the future. We then designed a tetracycline sensor system, because of the importance of antibiotics pollution in the light of multi-resistant pathogens. To this end, a tetracycline trigger plasmid was constructed by replacing the *araC* repressor gene and the ara operator of the arabinose trigger plasmid with the *tetR* gene coding for the tetracycline repressor and the tet operon. After combination with the memory plasmid, the tetracycline sensor system was shown to be functional in *E. coli* allowing to detect and memorize the presence of tetracycline. Due to a positive feedback between the trigger and memory systems, the combined whole-cell biosensor showed a very high sensitivity for tetracycline with a detection threshold at 0.1 ng/ml tetracycline, which may be a general property of sensors of this type. Moreover, acute presence of tetracycline and past exposure can be detected by this sensor using the dual readout of two reporter fluorophores.

## Introduction

*E. coli* is one of the best investigated organisms in biology and it is frequently used in the field of synthetic biology [1]. Its applications range from *E. coli* used to take high-definition chemical images [2], to cell-free transcription and translation systems consisting of an *E. coli* cytoplasmic extract [3]. Artificial genetic and epigenetic signaling networks in cells have been used to detect, process and store information about external stimuli in bacteria and mammalian cells [4–11]. In addition, DNA methylation based bistable gene expression switches have been described in bacteria that were based on two model well-studied DNA methyltransferases, the *Escherichia coli* deoxyadenosine DNA methyltransferase (Dam) and the *Caulobacter crescentus* cell-cycle-regulated methyltransferase (CcrM) [9,12]. In a previous work, we have developed a DNA methylation based epigenetic memory system in *E.coli* [9]. It is based on the CcrM DNA

252/35-1. The funders had no role in study design, data collection and analysis, decision to publish, or preparation of the manuscript.

**Competing interests:** The authors have declared that no competing interests exist.

methyltransferase and a designed zinc finger (ZnF) [13], which is capable to bind a target sequence overlapping with the recognition sequence of CcrM if it is unmethylated (Fig 1A). The tetrameric ZnF functions as repressor for the *ccrM* gene, which is combined with an enhanced green fluorescence (EGFP) reporter gene in a polycistronic operon. For gene

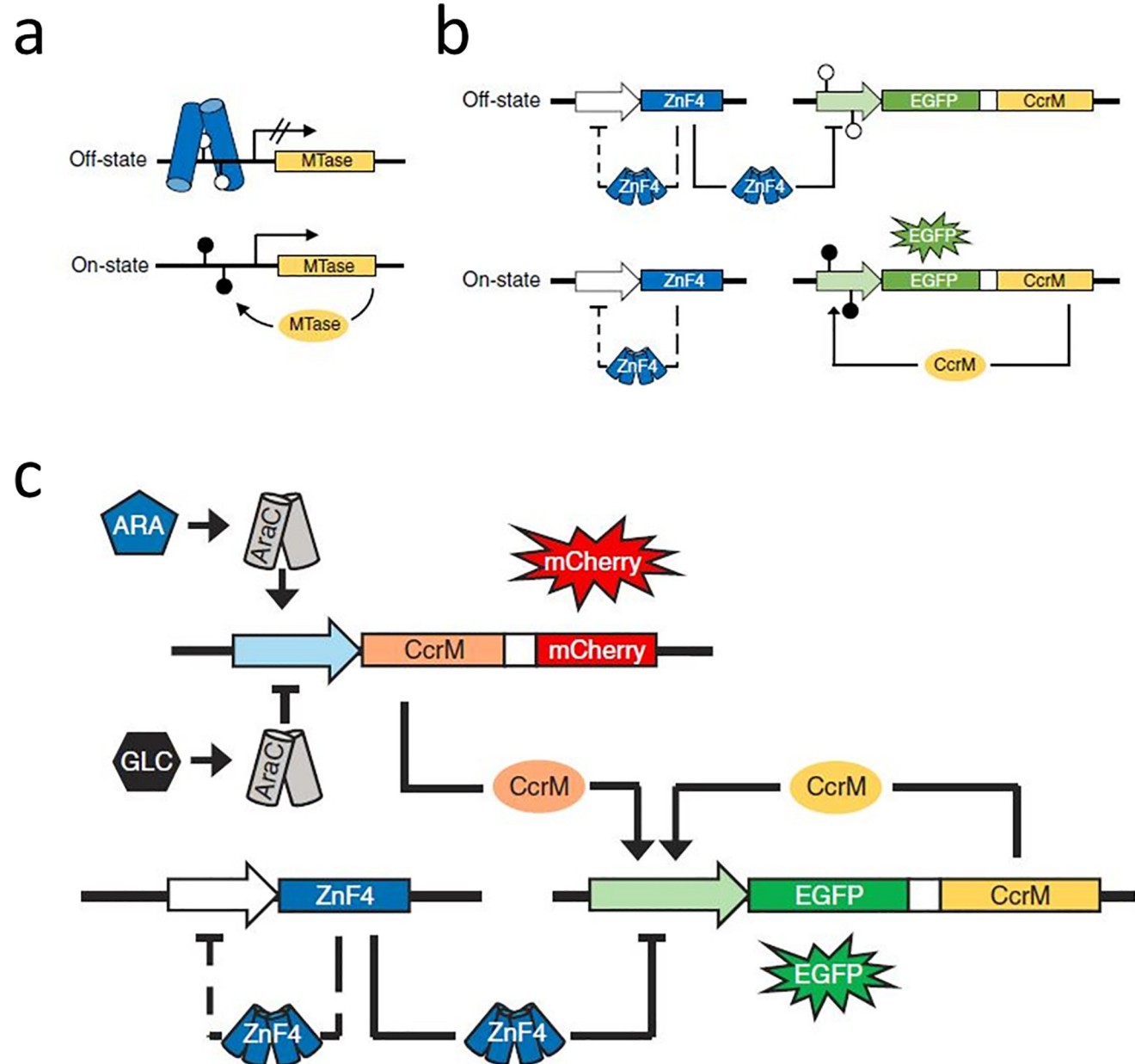

**Fig 1. Design of the arabinose memory system of Maier et al. (2017) [9].** a) General setup of the memory system in *E. coli*. Expression of a DNA methyltransferase (MTase) is repressed by a designed zinc finger that only binds if its palindromic target site is unmethylated (open lollipops). b) The tetrameric ZnF4 represses the memory plasmid operon encoding EGFP and CcrM and holds the systems in the OFF-state by binding to two palindromic binding sites. After an initial trigger, preventing that the ZnF4 can bind to the target sequence, CcrM is expressed and methylates the overlapping target sequence (filled lollipops). This leads to a positive feedback loop, holding the memory system in a trigger independent ON-state. In addition, ZnF4 represses its own expression in a methylation insensitive manner (dashed lines). c) Model of the arabinose memory system: L-arabinose is used as an initial trigger for the CcrM and mCherry expression, while glucose is repressing those genes. Once the initial CcrM is expressed by the arabinose trigger system, the memory system is switched on and holds a trigger independent ON-state because of the positive feedback loop. The figure panels were adopted with permission from Maier et al. (2017) [9].

regulation, double palindromic binding sites were used, which are overlapping with CcrM sites and hence can be methylated by CcrM. Under conditions when the target sequences are unmethylated, the gene expression is repressed (OFF-state of the memory system). The application of an external trigger causes the expression of a trigger CcrM from a trigger plasmid, which then methylates the memory plasmid (Fig 1B) [9]. This initial methylation causes a positive feedback loop, because the ZnF repressor dissociates from the methylated DNA leading to the loss of repression of the *ccrM* gene on the memory plasmid. Therefore, after the initial methylation, cells are switching into the ON-state, and CcrM is expressed from the memory plasmid independent of the trigger signal. This means that even if the initial trigger has disappeared the system holds its ON-state, which can be detected by the fluorescence of the reporter EGFP or by DNA methylation state analysis of the target sequence. Thereby, the information about the transient presence of the initial trigger is stored as methylation mark on the memory plasmid (Fig 1B) [9]. Maier et al. (2017) designed a trigger system for L-arabinose [9]. It contains a trigger plasmid encoding CcrM in an Ara operon, which switches from repression to expression if arabinose is present, whereas glucose is repressing those genes [14]. Combined with the memory system this leads to an arabinose memory system, which can store the information of the presence of arabinose in the medium in form of a stable EGFP signal (Fig 1C).

Tetracycline and its derivatives are well-known and widely used antibiotics in human and veterinary medicine, as animal growth promoters and for scientific usage [15]. With a production and consumption of thousands of tons per year, tetracycline antibiotics are the second most used antibiotics worldwide [16,17]. These antibiotics bind reversibly to the ribosomal 30S subunit and prevent binding of aminoacyl-tRNAs to the A-site of the 70S ribosome in a competitive manner. This reduces protein biosynthesis and causes the bacteriostatic effects of tetracycline [18]. Besides the great importance of tetracycline antibiotics in health care and the animal industry, they also have a disfavored impact on the environment [19]. About 30 to 90% of the antibiotics which were used for medication are released from humans and animals via urine and feces into the environment, because they were not absorbed or metabolized entirely [20–23]. In wastewater, for example, residual concentrations up to 2.37 µg/l tetracycline were detected even after wastewater treatment plants [24] and in ground waters near swine confinement facilities even higher concentrations of tetracycline were measured [25]. The high environmental pollution with antibiotics is critical, because it accelerates the spreading of multi-resistant pathogens [26,27].

There are several types of tetracycline resistance in bacteria, which exploit different mechanisms like export pumps, binding proteins, or proteins that prevent tetracycline binding to ribosomes [28]. They all have in common that they use a sensing system for tetracycline, which is able to detect the antibiotic at low concentrations and protect the organism against the antibiotic effects. Tight expression control of tetracycline resistance genes is necessary for their hosts, because expressing them needlessly would waste energy and they would have a negative impact on bacterial fitness [29]. The gene expression control of *tet*A encoding a tetracycline efflux gene by the tetracycline operon system of the Tn10 intrinsic region is well investigated and artificial constructs utilizing these sensing systems have found wide application for artificial gene regulation in all fields of biology and biotechnology for bacteria [30] as well as for eukaryotes [31,32]. The operator system contains the *tetR* gene coding for a homodimeric repressor protein, which binds two palindromic operator binding sites ($O_1$ and $O_2$). Both sites control the accessibility of the $P_{tetA}$ promotor, but also of $P_{tetR1}$ and $P_{tetR2}$ coding for the *tetR* repressor gene. In the absence of tetracycline, $O_1$ and $O_2$ are bound by TetR thus the expression of *tetR* and *tetA* is repressed and TetR is subject to autoregulation resulting in low but stable concentrations of TetR. If tetracycline is present, it binds to TetR homodimers, which leads

to a conformational change of TetR and its release from the operator binding sites. Once the accessibility to the promotors of *tetA* and *tetR* is given, both genes are expressed as long as enough tetracycline is in the cell to bind the repressor proteins. This leads to an inducible expression of the target gene TetA with a concentration dependent expression response and an efficient repression in the absence of tetracycline [33,34]. The TetR has been shown to bind efficiently to several chemically modified forms of tetracycline as well, including pharmacologically relevant forms and degradation products [30,35,36].

In this work we have adapted and validated a flow cytometry readout for the arabinose sensor system established by Maier et al. (2017) which is more convenient and enables a multiparameter analysis and single cell readout [9]. Based on this, we then designed an antibiotic detection sensor system using tetracycline as an example, because of the importance of antibiotics pollution in the light of multi-resistant pathogens. To this end, a new tetracycline trigger plasmid was constructed by replacing the *araC* repressor gene and the ara operator of the arabinose trigger plasmid with the *tetR* gene coding for the tetracycline repressor and the tet operon. The sensor system was shown to be functional in *E. coli* allowing to detect and memorize the presence of tetracycline. Due to the positive feedback between the trigger and memory systems, the combined biosensor showed a very high sensitivity for tetracycline, which may be a general property of whole-cell biosensors of this type.

## Experimental section

### List of plasmids taken from Maier et al. (2017)

All plasmids used in this work are compiled in S1 Text.

### Cloning of the trigger plasmid responding to tetracycline

Starting from the arabinose trigger plasmid [9], the pBAD promoter together with the *araC* gene was replaced by the tetR/tetA promoter and the tetracycline repressor gene *tetR* (3420–4127 bp) taken from the pC008-pBR322 plasmid containing a *tetR*-inducible red fluorescent protein (Addgene 79157) using Gibson assembly (NEB) with following primers:

FP TriTet RBS Insert `ATTGTCTGATTCGTTACCAATTAAGACCCACTTTCACATTTAAGTTGTTTTTCTAATCC`

RP TriTet RBS Insert `GCTAGCCCAAAAAAACGGGTTTCTCTATCACTGATAGGGAGTGGTAAAATAACTCTATCA`

FP TriTet RBS Vector `TCCCTATCAGTGATAGAGAAACCCGTTTTTTTGGGCTAGC`

RP TriTet RBS Vector `AATGTGAAAGTGGGTCTTAATTGGTAACGAATCAGACAATTGACGGC`

The annotated sequence of the tetracycline trigger plasmid is provided in S2 Text.

### Measurement of the toxicity of tetracycline

Using a pre-culture of DH5α *E. coli* cells in LB medium without tetracycline, LB cultures containing different concentrations of tetracycline were inoculated with an $OD_{600\,nm}$ of 0.05, incubated with shaking at 37˚C and the $OD_{600\,nm}$ regularly determined during the exponential growth phase. All experiments were carried out in triplicate.

### Tetracycline induced gene expression

DH5α electro competent *E. coli* cells were co-transformed with 5 ng memory system plasmid and 5 ng tetracycline trigger plasmid. As negative control, 5 ng negative control memory plasmid containing a catalytically inactive D31A CcrM mutant, was used. Cells were cultured for 2

h at 28˚C in SOC medium supplemented with 30 µM $ZnSO_4$. Cells were then plated on LB agar plates containing 25 µg ml$^{-1}$ kanamycin, 75 µg ml$^{-1}$ ampicillin, 30 µM $ZnSO_4$ and 0.2% glucose and incubated at 28˚C for 30 to 48 h. Single colonies were used to inoculate overnight cultures in LB medium containing 25 µg ml$^{-1}$ kanamycin and 100 µg ml$^{-1}$ ampicillin, 10 or 30 µM $ZnSO_4$ cultivated at 28 or 29˚C. Expression of the *ccrM* gene in liquid culture was induced by addition of tetracycline (0.1 ng/ml to 10 ng/ml).

### Arabinose induced gene expression

Experiments with the arabinose induced gene expression system were basically conducted as described [9]. Briefly, XL1-Blue electro competent *E. coli* cells were co-transformed with the memory plasmid or the negative memory plasmid and arabinose trigger plasmid (5 ng each). Cells were cultured for 2 h at 28˚C in SOC medium supplemented with 30 µM $ZnSO_4$. Cells were then plated on LB agar plates containing 25 µg ml$^{-1}$ kanamycin, 75 µg ml$^{-1}$ ampicillin, 30 µM $ZnSO_4$ and 0.2% glucose and incubated at 28˚C for 30 to 48 h. Single colonies were used to inoculate overnight cultures in LB medium containing 25 µg ml$^{-1}$ kanamycin and 100 µg ml$^{-1}$ ampicillin, 10 µM $ZnSO_4$ and 0.2% glucose cultivated at 28˚C. Expression of the *ccrM* gene in liquid culture was induced by exchange of the glucose containing to arabinose (0.02% (w/v)) containing medium.

### Whole-cell lysate fluorescence and cell growth measurements for the detection of the reporter gene expression

Whole-cell lysate fluorescence measurements were conducted as previously described [9]. Cell density was normalized by the $OD_{260\ nm}$ of the bacterial cell lysates.

### Flow cytometry measurements for the detection of the reporter gene expression

To avoid contamination of the flow cytometer with live bacteria, the *E. coli* cells were washed and afterwards treated with paraformaldehyde (PFA) to fix them before flow cytometry measurements. Treated and untreated samples were incubated overnight on separate agar plates to validate that no colony-forming units were present in the treated samples in contrast to the untreated cells. In detail, bacterial cultures were grown until an $OD_{600\ nm}$ of ~1.0. About 200 µl sample was harvested by centrifugation for 2 min at 11,000 rpm. For bacterial cell fixation the supernatant was discarded and the pellet was resuspended in 1.5 ml Z-Buffer (60 mM $Na_2HPO_4$, 40 mM $NaH_2PO_4$, 10 mM KCl, 1 mM $MgSO_4$ adjusted to pH 7 and sterile filtered with a 0.2 µM filter (Filtropur BT25, Sarstedt)) and centrifuged for 2 min at 11,000 rpm. Afterwards, the cell pellet was resuspended in 600 µl 1% (w/v) PFA (Paraformaldehyde EM Grade, 00380–1, Polysciences, prepared in Z-Buffer) and incubated for 15 min at room temperature. After centrifugation for 3 min at 13,000 rpm, the pellet was resuspended in 600 µl Z-buffer and filtered through 30 µm Pre-Separation Filters (Miltenyibiotec) into 5 ml Polystyrene Round-Bottom Tubes (BD Falcon). Expression of the EGFP and mCherry reporter genes in *E. coli* cells was evaluated for approximately 30,000 events via flow cytometry (MACSQuant® VYB Flow Cytometer, Miltenyi Biotec) using standard settings (Fig 2A). All flow cytometry data were evaluated with FlowJo V10 software (FlowJo™ Becton, Dickinson and Company). Gates for EGFP and mCherry were optimized to allow most stringent discrimination between ON- and OFF-state cells of the different sensor systems and assay temperatures and used for all experiments under one set of conditions.

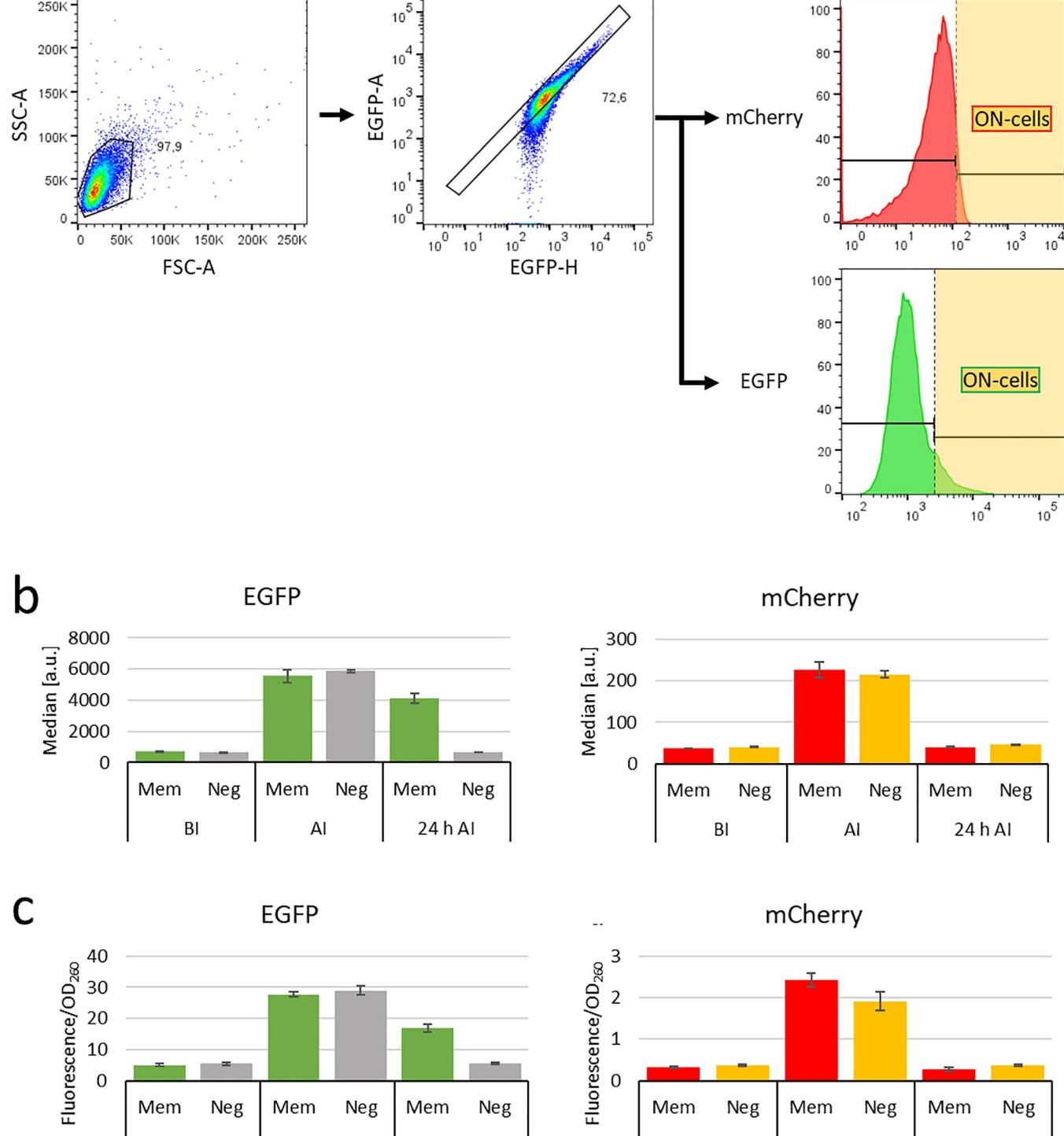

**Fig 2. Establishment of a flow cytometry readout of the arabinose memory system.** a) Schematic illustration of the principle flow cytometry analysis workflow. The *E. coli* cells are selected with the first gate on the basis of their forward scatter area (FSC-A) and sideward scatter area (SSC-A). The single cells of the *E. coli* population are selected with a second gate on the diagonal of an EGFP area (EGFP-A) to EGFP height (EGFP-H) distributions. This single cell population is further analyzed for fluorescence area in normalized histograms with a third gate which selects cells with a certain fluorescence level to be in ON-state as illustrated in S2A Fig. b) Flow cytometry signals of the arabinose sensor system. *E. coli* containing the arabinose trigger plasmid and either the memory

plasmid (Mem) or the negative control memory plasmid with a catalytically inactive CcrM mutant (Neg) were grown overnight at 28°C either in glucose containing medium (before induction, BI) or arabinose containing medium (after induction, AI). Afterwards, the induced cultures were grown in glucose containing medium for further 24 h (24 h AI). c) Whole-cell lysate fluorescence signals of the same samples as shown in panel b. The EGFP signal of the memory system is shown in green bars, EGFP signal of the negative control in gray bars, mCherry signal of the memory system in red bars, mCherry signal of the negative control in orange bars. All averages are based on 3 biological replicates, error bars display the SD.

## Results and discussion

### Establishment of flow cytometry detection for the arabinose memory system

Flow cytometry is a widely used technique to investigate all kind of particles with respect to their shape, granularity and fluorescence. The size of the particles, which can be analyzed, ranges from mammalian cells to much smaller bacteria cells [37,38]. This technology has been successfully used in the eukaryotic and prokaryotic sensors mentioned above and it was our first aim to establish the routine application of flow cytometry for the quantitative readout of our epigenetic memory systems, which previously was mainly based on fluorescence spectroscopy in whole-cell lysates [9]. To this end, *E. coli* cells were grown in a glucose-containing medium at 28°C and analyzed before induction and after induction when glucose in the medium was exchanged by arabinose. For stability investigations, the samples were grown after induction for further 24 h under the same conditions as before induction (i.e. without arabinose and in presence of glucose). As a negative control (Neg), a system containing a catalytically inactive D31A CcrM mutant in the memory plasmid but active CcrM in the trigger plasmid was used. Afterwards, the samples were analyzed with flow cytometry and the established whole-cell extract fluorescence spectroscopy technique side-by-side to compare the results of both methods (Fig 2).

Initially we measured the mCherry signal indicating expression of the trigger plasmid in the memory system and the negative control system (red or orange bars) with flow cytometry and fluorescence spectroscopy. As shown in Fig 2, both signals increased after induction and immediately decreased after 24 h to the same level as before induction. In contrast, the medians of the EGFP signal indicating the memory system expression (green bars) measured with flow cytometry and the fluorescence spectroscopy measurements showed a clear memory effect 24 h after induction, because the EGFP signal was still detectable, but the mCherry signal corresponding to the trigger plasmid expression has been lost. As expected, the EGFP signal of the negative control (gray bars) demonstrated no memory effect, because the inactive CcrM cannot propagate the methylation signal. Overall, the flow cytometry and fluorescence spectroscopy readout showed a very comparable dynamic behavior and similar relative signal changes. This indicates that flow cytometry provides reliable data for the arabinose memory system.

As mentioned before, the flow cytometry readout allows for a single cell analysis of the data. To exploit this option, the overall cell population was split into an OFF-state and an ON-state population using gates for EGFP and mCherry that were optimized to discriminate cells before and after induction (Fig 3). With this ON-state cell analysis, a better discrimination of the EGFP and mCherry signals was achieved, and the memory effect could be documented more clearly. This type of ON-state cell analysis is only possible with the single cell analysis provided by flow cytometry and it represents a major advantage of this method compared to the readout by whole-cell lysate fluorescence spectroscopy. The ON-state cell analysis is reasonable especially in the perspective to use different memory systems in a combined way to generate Boolean logic gates which need a binary OFF or ON (0 or 1) input and output [39]. Another advantage of the flow cytometry readout system is its more convenient handling, because no

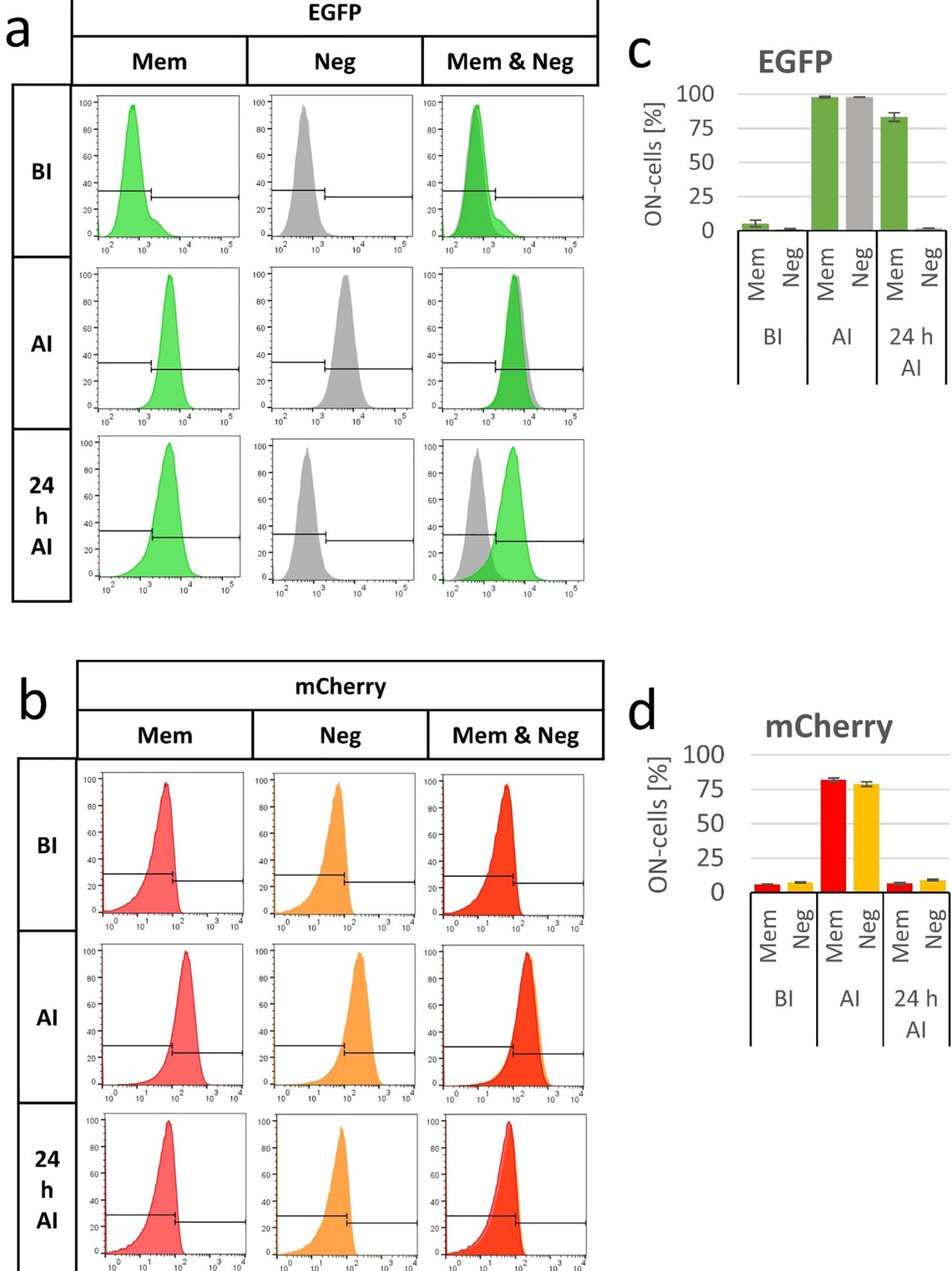

**Fig 3. ON-state cell analysis of the arabinose memory system.** a) and b) Exemplary flow cytometry histograms of the arabinose memory system (Mem) or the negative control (Neg) of the EGFP (a) and mCherry (b) signals. The gates which were set to distinguish between ON- and OFF-state cells are indicated. All events on the right side of the gate were defined as ON-state cells. The third column displays the Mem and Neg distributions in one histogram to allow better comparison. The x-axis displays the fluorescence intensity, the y-axis the fraction of cells normalized to the highest value. c) and d) percentage of ON-state cells (ON-cells) for EGFP (c) and mCherry (d) before induction (BI), after induction (AI) and 24 hours after induction (24 h AI). The EGFP signal of the memory system is shown in green bars, EGFP signal of the negative control in gray bars, mCherry signal of the memory system in red bars, mCherry signal of the negative control in orange bars. Averages are based on 3 biological replicates, error bars display the SD. Experiments were conducted at 28˚C.

cell lysis is necessary and no calibration of the cell number is needed, which was done in the fluorescence readout by a parallel absorption measurement of the cell lysate at 260 nm. These handling advantages made the experiment procedure faster and more consistent and, thereby, increased the quality of the collected data.

## Design of a tetracycline sensor system

Tetracycline is a well-known and widely used antibiotic in human and veterinary medicine, as animal growth promoter and in basic research applications [15]. However, because of the regular application in these fields, it increasingly pollutes the environment [19]. Therefore, we aimed to design a cellular tetracycline sensor in *E. coli* featuring memory functions. The new tetracycline trigger system was designed in a way that the already existing memory plasmid [9] could be used without alterations (Fig 4A). Therefore, a new tetracycline inducible trigger plasmid (TriTet) was constructed featuring CcrM and mCherry expression on a polycistronic operon under the control of a tet operator system (Fig 4B). The new TriTet plasmid can be connected to the memory system in the same way as in the arabinose memory system. In the absence of tetracycline, CcrM and mCherry expression of the trigger plasmid is repressed by the tet-repressor. In the presence of tetracycline, the tet-repressor is released from the DNA and TriTet expresses the trigger CcrM and mCherry. Then, the trigger CcrM methylates the promotor region of the memory plasmid *ccrM* gene, which afterwards establishes a positive feedback loop leading to stable expression of EGFP and CcrM.

## Experimental validation of the new tetracycline sensor system

The newly designed trigger plasmid was tested in combination with the memory system. For all experiments, the *E. coli* strain was changed from XL1-Blue to DH5α, because XL1-blue expresses a tetracycline efflux system [18,40], which was not desirable for the experiments. To determine a suitable working concentration of tetracycline, we first measured the sensitivity of DH5α cells to this antibiotic. First, it was confirmed that the DH5α cells were unable to grow on LB plates containing normal concentrations of tetracycline (10 μg/ml). Next, we cultivated DH5α cells in liquid LB cultures containing different concentrations of tetracycline. Measurement of cell division rates during the exponential growth phase at 37˚C indicated an absence of a detectable effect of tetracycline on cell growth up to concentrations of 30 ng/ml (S1 Fig). Based on this, we used tetracycline concentrations up to 10 ng/ml in our study.

Initial tests to optimize the tetracycline memory system indicated that an optimum compromise of response strength, low leakiness and stability of switching was observed with liquid medium containing 30 μM ZnSO$_4$ and if cells are cultivated at 29˚C. After induction of the system with 10 ng/ml tetracycline, the EGFP and mCherry signals increased (Fig 5). 24 h after induction, a clear memory effect was detectable in comparison of the memory system with the transient signal increases of EGFP in the negative control and the mCherry signal. The same experiment was conducted at 28˚C with roughly the same results (S2 Fig). These data indicate

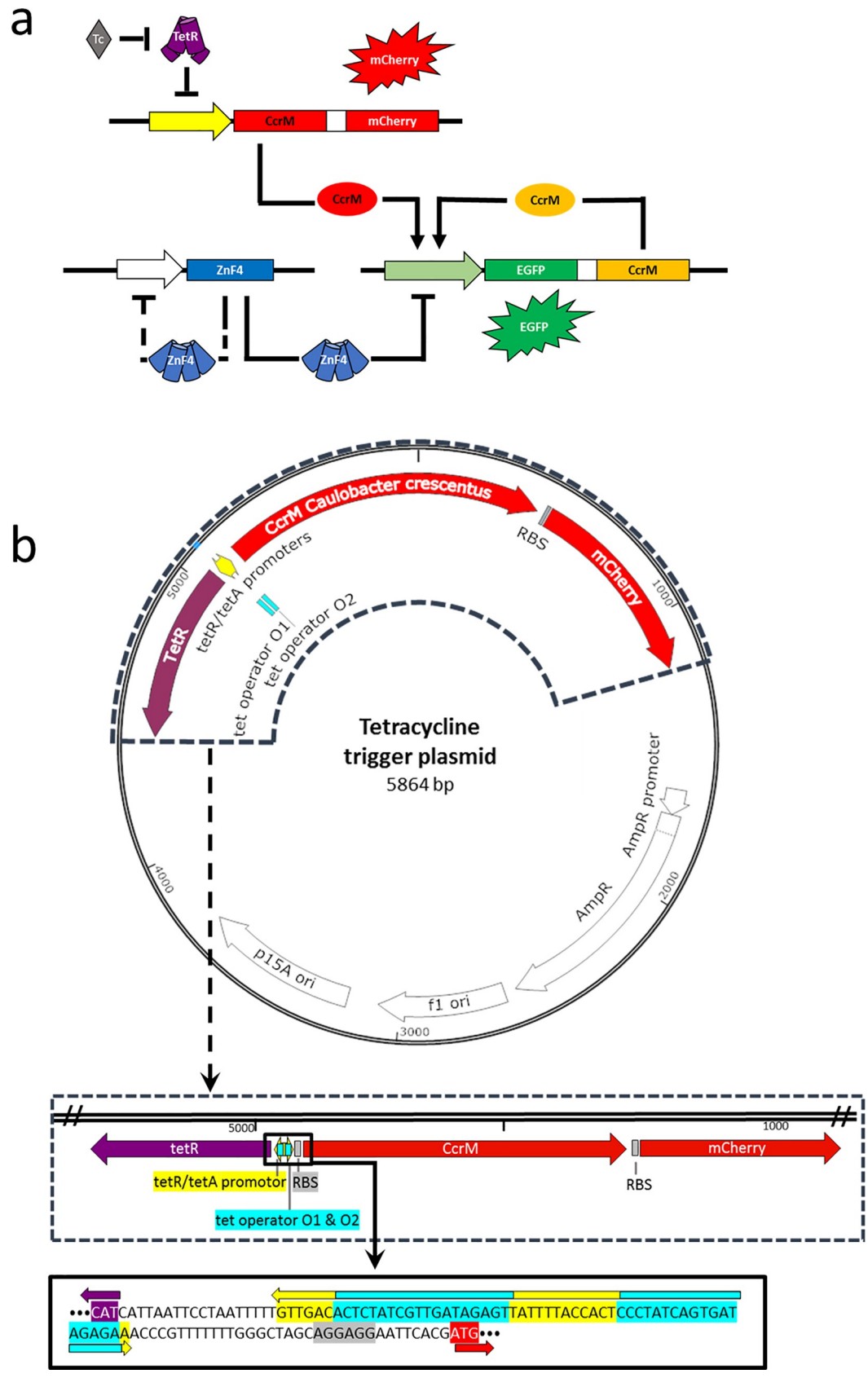

**Fig 4. Design of the tetracycline memory system.** a) Model of the tetracycline memory system: Tetracycline (Tc) is used as an initial trigger for the CcrM and mCherry expression. Once the initial CcrM is expressed by the tetracycline trigger system the memory system is switched on and holds a trigger independent ON-state because of a positive feedback loop. b) Design of the tetracycline trigger plasmid, which was constructed based on the existing arabinose trigger plasmid [9]. Therefore, a tetracycline inducible operator (tet operator, cyan), tetR/tetA promotors (yellow) and the tetracycline repressor gene *tetR* (purple) were utilized to achieve a tetracycline inducible expression of the *ccrM* gene (red) combined with an mCherry reporter gene in a polycistronic manner. The lower parts show magnifications of the TetR memory elements containing regions, the promoter and operator sequences. RBS, ribosome binding site. The sequence of the tetracycline trigger plasmid is provided in S2 Text.

that the new trigger plasmid can be successfully integrated into the established epigenetic memory system.

In summary, a tetracycline responsive trigger plasmid was designed and integrated into the existing memory system. Flow cytometry revealed that this plasmid could be triggered with tetracycline and it was used to generate a functional tetracycline memory system. However, the tetracycline memory system still showed a less stable OFF-state than the arabinose memory system. Given that the memory parts of both systems are identical, this result suggests that the CcrM promoter in the tetracycline trigger plasmid has a greater leakiness than the corresponding element in the arabinose trigger plasmid, which may be improved in future design.

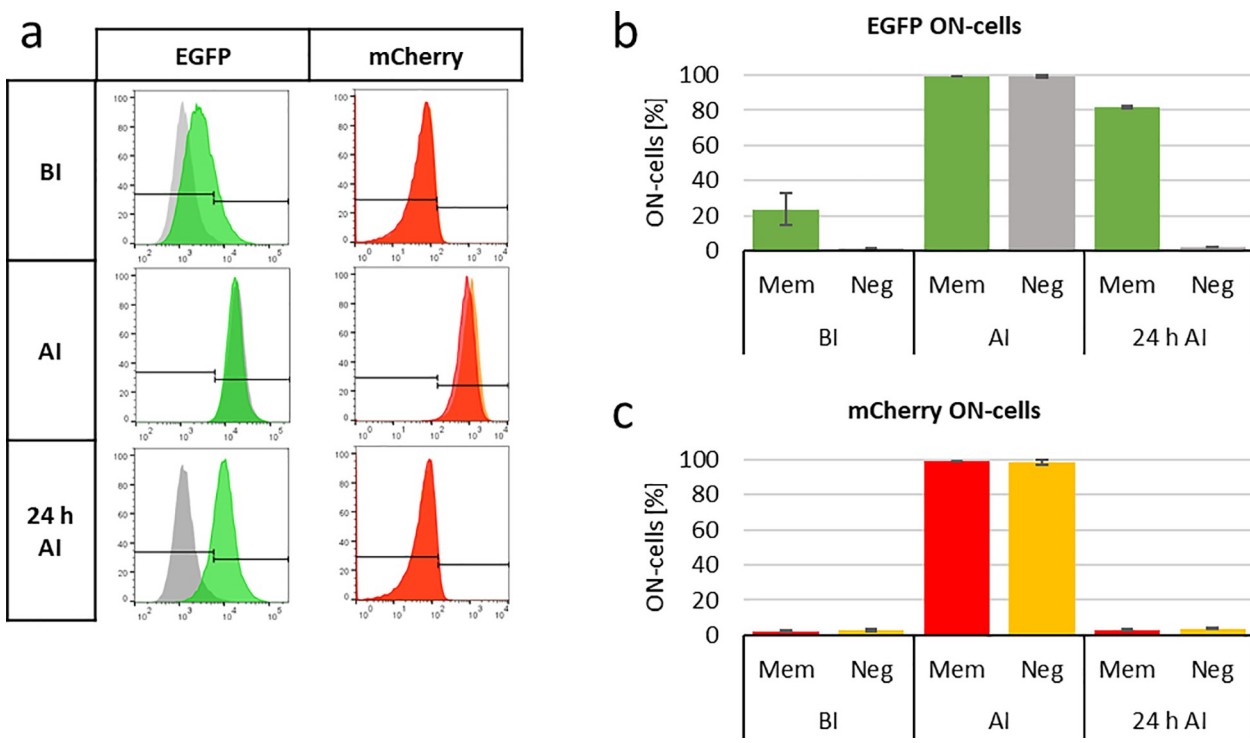

**Fig 5. Investigation of the tetracycline sensor system.** EGFP (green) and mCherry (red) fluorescence was measured by flow cytometry of *E. coli* carrying the trigger plasmid and either the memory plasmid (Mem) or the negative control memory plasmid with an inactive CcrM mutant (Neg). Both were cultured at 29°C with 30 μM $ZnSO_4$ without tetracycline (before induction, BI) and with 10 ng/ml tetracycline (after induction, AI). The induced cultures were grown for further 24 h in a medium without tetracycline (24 h AI). a) Histograms of the single cell populations and the applied gates. b) and c) Percentage of ON-state cells (ON-cells) for EGFP and mCherry. The EGFP signal of the memory system is shown in green bars, EGFP signal of the negative control in gray bars, mCherry signal of the memory system in red bars, mCherry signal of the negative control in orange bars. Averages are based on 3 biological replicates, error bars indicate the SD. For a corresponding experiment conducted at 28°C and 10 μM $ZnSO_4$, see S2 Fig.

## Application of the tetracycline sensor system as whole-cell biosensor

Next, the newly designed tetracycline memory system was applied as an *E. coli* whole-cell bio-sensor to sense tetracycline, because tetracycline is present in the environment and a whole-cell biosensor with a memory effect could be used for tetracycline monitoring. Whole-cell bio-sensors are genetically engineered microorganisms which were designed to respond in a dose-dependent manner to changes in environmental conditions [41]. The general idea is that an extracellular signal from the environment is converted *in vivo* by the genetically engineered organism into a signal which can be read by a technical device. In the case of the tetracycline memory system whole-cell biosensor, tetracycline represents the extracellular signal, which is converted into two different fluorescence signals. The primary mCherry signal from the trigger plasmid reflects the acute presence of tetracycline and the secondary EGFP signal from the memory plasmid reflects an exposure in the past. Both signals can be read out by flow cytometry or fluorescence spectroscopy.

To test the function and sensitivity of our tetracycline whole-cell sensor, *E. coli* cells carrying the tetracycline sensor system, were cultivated at 29˚C and induced for 24 h with several tetracycline concentrations. The induced cultures were grown for further 21 h in a medium without tetracycline and all samples were analyzed by flow cytometry (Fig 6 and S3 Fig). After induction, the ON-state cell analysis of the EGFP and the mCherry signal demonstrated signal increases from 0 ng/ml tetracycline to 10 ng/ml tetracycline for the memory system and the

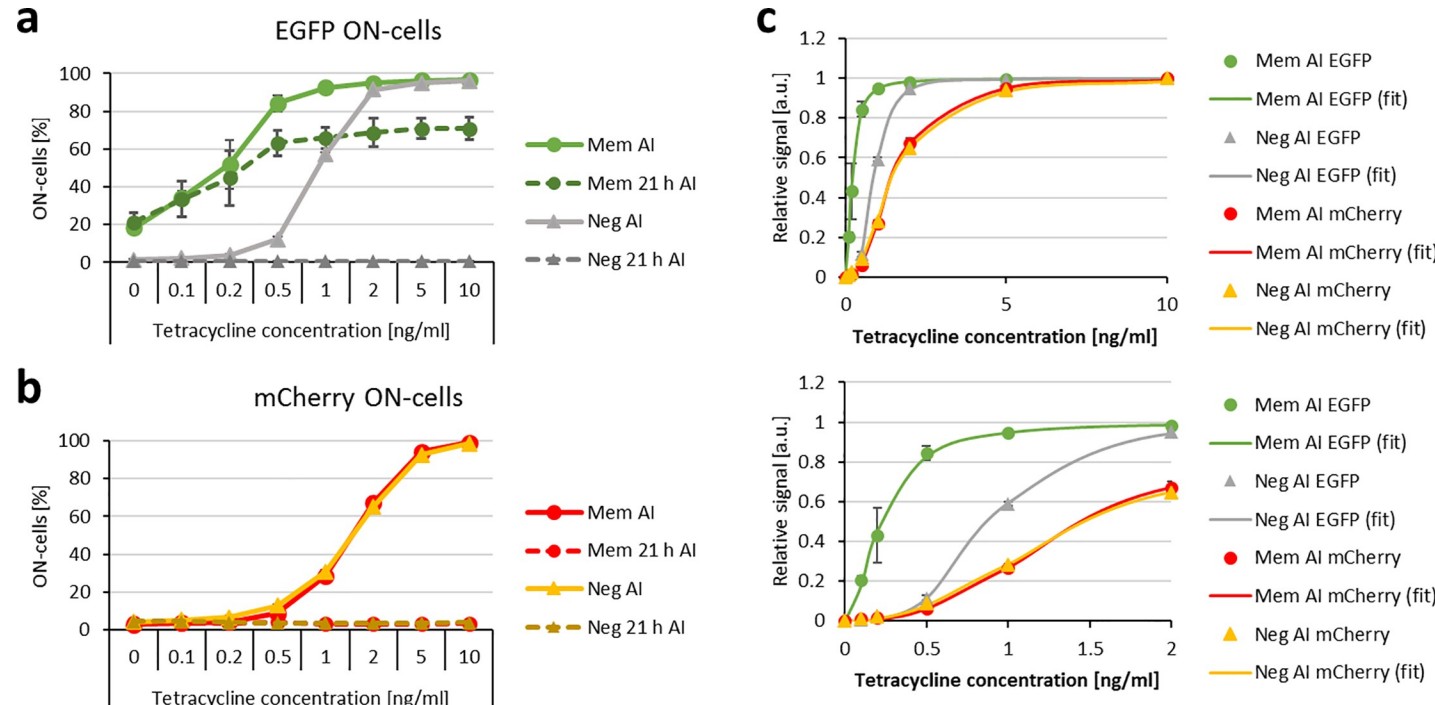

**Fig 6. Sensitivity of the tetracycline sensor system.** EGFP and mCherry signals were measured with flow cytometry of *E. coli* carrying the tetracycline trigger plasmid and either the memory plasmid (Mem, circles) or the negative control memory plasmid (Neg, triangles). The cells were cultured with 30 μM ZnSO₄ at 29˚C without tetracycline (before induction, BI) and induced with up to 10 ng/ml tetracycline (after induction, AI). The induced cultures were grown for further 21 h in medium without tetracycline (21 h AI). a) and b) Percentage of ON-state cells (ON-cells) shown for EGFP and mCherry. Histograms of the single cell populations and the applied gates are shown in S3 Fig. c) Fraction of ON-state cells based on the EGFP and mCherry fluorescence signals after induction normalized to the fraction of ON-state cells at 10 ng/ml tetracycline. The lines show fits of the data to the Hill-equation revealing apparent half saturation levels of 0.2 and 0.9 ng/ml tetracycline for the EGFP signals of the memory and negative control system and 1.5 ng/ml tetracycline for the corresponding mCherry signals. The lower image shows an enlargement of the x-axis. Error bars always indicate the SD of 3 biological replicates of the memory system and 2 biological replicates of the negative control system. Note that error bars sometimes are smaller than the data points, hence they are not visible.

negative control system, which carried a catalytically inactive CcrM mutant instead of the wildtype CcrM. Strikingly, for at least 0.5 ng/ml tetracycline and all higher concentrations, the information of the transient tetracycline presence was stored for at least 21 h after the stimulus was removed as indicated by a significant EGFP signal increase as compared to the uninduced samples (S1 Table).

## The tetracycline whole-cell sensor features a very high sensitivity

As shown in Fig 6C, a strong signal enhancing effect caused by the memory system was observed. This effect is apparent in the concentration dependent responses of the memory system and the negative control to different tetracycline concentrations. The mCherry trigger signal of the memory system and the negative control behaved the same way with an apparent half saturation at about 1.5 ng/ml tetracycline. However, the EGFP signals of the memory system and the negative control both showed a response at lower tetracycline concentrations than the mCherry trigger signal. For the negative control system, the apparent half saturation was observed at 0.9 ng/ml tetracycline and for the functional memory system, a response at even lower concentrations was observed, with an apparent half saturation at 0.2 ng/ml tetracycline. With this system even at 0.1 ng/ml tetracycline, a significant increase of the EGFP signal compared to the uninduced samples was observed (S1 Table).

These findings indicate that for low tetracycline concentrations the secondary EGFP memory signal was enhanced in two steps in comparison to the primary mCherry trigger signal. One amplification step was due to the transfer of the trigger signal via the trigger CcrM to the memory plasmid or negative control memory plasmid, because one CcrM molecule can methylate and thereby de-repress several plasmids and these can generate many EGFP mRNAs. The second amplification step is caused by the positive feedback loop of memory system in comparison to the negative control. This effect is due to the active CcrM that is produced from the memory plasmid and enhances the effect of the trigger CcrM. These signal enhancing effects cause a very sensitive detection of tetracycline by the whole-cell sensor that could be also beneficial for other whole-cell biosensors in applications which require a high sensitivity. One disadvantage of the high sensitivity and switch like behavior of the system is that it is less suitable to determine actual concentrations but optimized to detect the presence of the analyte at concentrations above the detection threshold.

The tetracycline whole-cell biosensor generated in this work was able to detect low tetracycline concentrations and a memory effect was shown for at least one day. A signal enhancing effect inherent in the memory system was observed allowing this new whole-cell biosensor in *E. coli* to reach a detection threshold of 0.1 ng/ml tetracycline which is well below of tetracycline concentrations observed in the environment [24,25]. This sensitivity is high when compared with other reports. Using a β-galactosidase reporter system a limit of detection (LOT) of 1 ng/ml tetracycline was achieved and with a fluorescence reporter system, an LOT of about 10 ng/ml was reported [42]. Improved versions of this system as whole-cell biosensors were able to sense 5 ng/ml up to 16 μg/ml tetracycline [35]. Another *E. coli* based tetracycline whole-cell biosensor demonstrated a LOT of 50 ng/ml oxytetracycline [43]. In summary, the tetracycline detection system developed here combines the multiple turnover and positive feedback loop of CcrM to the expression of EGFP to establish a highly sensitive live cell tetracycline biosensor.

## Conclusions

Bacterial live cell sensors are potentially powerful tools for the detection of environmental toxins which can be equipped with a memory potential using an epigenetic DNA methylation based system. In this work, we expanded an existing DNA methylation based arabinose sensor

with memory function by designing a tetracycline sensor input system. After introduction into *E. coli*, the system was able to detect and memorize the presence of tetracycline with a very high sensitivity, which originated from a positive feedback between the trigger and memory systems. Moreover, acute presence of tetracycline and past exposure can be detected and distinguished by this sensor using the dual readout of two reporter fluorophores. After incorporation into an appropriate immobilization device [44,45], *E. coil* cells containing this system could constitute a highly sensitive whole-cell biosensor for tetracycline detection in liquid samples compatible with bacterial cultivation. In order to increase genetic stability of the biosensor in bacteria and avoid potential fluctuations in plasmid copy numbers under various growth condition, the sensor components could be integrated into the bacterial genome. Readout could be achieved by flow cytometry, fluorimeters, alternatively by UV-spectroscopy or with chromogenic substrates when using other reporter proteins. Moreover, acute presence of tetracycline and past exposure can be detected by one sensor making it a powerful system for the detection of tetracycline and long-term surveillance of environmental sites.

## Supporting information

**S1 Table. Statistical analysis of the readout of the tetracycline memory system used as whole-cell biosensor shown in Fig 6.**
(PDF)

**S1 Fig. Sensitivity of DH5α cells to tetracycline in liquid cultures.**
(PDF)

**S2 Fig. Investigation of the tetracycline sensor system at 28°C with 10 μM ZnSO4.**
(PDF)

**S3 Fig. ON-state cell analysis of the tetracycline sensor system.**
(PDF)

**S1 Text. List of all plasmids used in this paper.**
(PDF)

**S2 Text. Tetracycline trigger plasmid annotated DNA sequence.**
(PDF)

## Author Contributions

**Conceptualization:** Sara Weirich, Albert Jeltsch.

**Formal analysis:** Timo Ullrich, Sara Weirich, Albert Jeltsch.

**Funding acquisition:** Albert Jeltsch.

**Investigation:** Timo Ullrich, Sara Weirich.

**Methodology:** Timo Ullrich, Sara Weirich.

**Project administration:** Albert Jeltsch.

**Resources:** Albert Jeltsch.

**Supervision:** Sara Weirich, Albert Jeltsch.

**Validation:** Timo Ullrich, Sara Weirich.

**Visualization:** Timo Ullrich, Sara Weirich, Albert Jeltsch.

**Writing – original draft:** Timo Ullrich, Sara Weirich, Albert Jeltsch.

**Writing – review & editing:** Timo Ullrich, Sara Weirich, Albert Jeltsch.

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
