## [Decision Letter · Decision Letter 0]

10 Apr 2020

PONE-D-20-06852

Development of an epigenetic tetracycline sensor system based on DNA methylation

PLOS ONE

Dear Prof. Dr. Jeltsch,

Thank you for submitting your manuscript to PLOS ONE. After careful consideration, we feel that it has merit but does not fully meet PLOS ONE’s publication criteria as it currently stands. Therefore, we invite you to submit a revised version of the manuscript that addresses the points raised during the review process.

Both reviewers are in agreement that this work is technically sound with data supporting the conclusions and used the appropriate statistics. Thus we have made an editorial decision of minor revisions needed. In your revisions, please address Reviewer #2's suggestions/questions as well as considering the suggested writing edits to this manuscript. Please let me know should you have any questions about these revisions.

We would appreciate receiving your revised manuscript by May 25 2020 11:59PM. To enhance the reproducibility of your results, we recommend that if applicable you deposit your laboratory protocols in protocols.io, where a protocol can be assigned its own identifier (DOI) such that it can be cited independently in the future. For instructions see: http://journals.plos.org/plosone/s/submission-guidelines#loc-laboratory-protocols

We look forward to receiving your revised manuscript.

Kind regards,

Nathaniel A. Hathaway, Ph.D.

Academic Editor

PLOS ONE

Journal Requirements:

Reviewers' comments:

Reviewer's Responses to Questions

**Comments to the Author**

1. Is the manuscript technically sound, and do the data support the conclusions?

Reviewer #1: Yes

Reviewer #2: Yes

2. Has the statistical analysis been performed appropriately and rigorously? 

Reviewer #1: Yes

Reviewer #2: Yes

3. Have the authors made all data underlying the findings in their manuscript fully available?

Reviewer #1: Yes

Reviewer #2: Yes

4. Is the manuscript presented in an intelligible fashion and written in standard English?

Reviewer #1: Yes

Reviewer #2: Yes

5. Review Comments to the Author

Reviewer #1: In the manuscript, Ullrich and co-authors develop a highly sensitive flow cytometery analysis and tetracycline biosensor in E. Coli. This new sensor platform was modified from a previous arabinose biosensor/memory system, applying a design of genetic circuit consisted of a triggering plasmid responding to tetracycline and a memory plasmid that establish memory after triggering signal was removed by controlling DNA methylation state on memory plasmid. Combining the signal enhancing effects from the circuit design and the high sensitivity of flow cytometery, this new system can push the detection limit of tetracycline to 0.1 ng/mL, which is much lower than previously reported systems. Overall, this manuscript was well written, the experiments carefully designed and performed, and the conclusions were supported by the data. This work should be of interest to the readership of PLOS ONE and I recommend to accept this manuscript for publication.

Reviewer #2: Suggestions and questions:

1. Tetracycline is degraded by by ultraviolet radiation, and also (less efficiently) by visible light. As a consequence, reliable monitoring of tetracycline in the environment should ideally detect chemical forms derived from tetracycline. Such forms have usually lost antibiotic activity. To mimic this situation, the authors might test whether the biosensor described in this manuscript is able to detect autoclaved chlortetracycline, which has inducing activity but not antibiotic activity.

2. Papers published in the 1980's (e. g., Moyed et al. 1983) showed that overexpression of tetracycline resistance proteins (e. g., from a multicopy plasmid) reduce tetracycline resistance. Should this phenomenon be taken into account? How important is plasmid copy number to make sure that the tetracycline sensor will work?

Comments about writing (line numbers are tentative and have been identified by the reviewer):

3. The introduction is very long. The description of the arabinose biosensor that provided the background for this study could be shortened without loss of clarity (especially considering that a figure describing the sensor is provided).

4. Page 3, line 6. Because epigenetic biosensors are rare in bacteria, you might give some more details about, for instance indicating that the previously described sensors (refs. 9 and 12) are based on DNA methylation by two model DNA methyltransferases, CcrM and Dam.

5. Page 2, line 14. Remove "the" before "CcrM".

6. Page 7, first paragraph. Flow cytometry has been widely used in eukaryotic epigenetic sensors and also in the prokaryotic sensor described in ref. 12. You might mention this fact to further support your strategy in this study (and also to be fair to the literature).

7. Page 11, Conclusions. Please revise the sentence that starts "This work...". The current sentence is difficult to read!

6. PLOS authors have the option to publish the peer review history of their article (what does this mean?). If published, this will include your full peer review and any attached files.

Reviewer #1: No

Reviewer #2: Yes: Josep Casadesús

---

## [Author Response · Author response to Decision Letter 0]

13 Apr 2020

PONE-D-20-06852

Development of an epigenetic tetracycline sensor system based on DNA methylation

Response to the reviewers’ comments

Reviewer #1: In the manuscript, Ullrich and co-authors develop a highly sensitive flow cytometery analysis and tetracycline biosensor in E. Coli. This new sensor platform was modified from a previous arabinose biosensor/memory system, applying a design of genetic circuit consisted of a triggering plasmid responding to tetracycline and a memory plasmid that establish memory after triggering signal was removed by controlling DNA methylation state on memory plasmid. Combining the signal enhancing effects from the circuit design and the high sensitivity of flow cytometery, this new system can push the detection limit of tetracycline to 0.1 ng/mL, which is much lower than previously reported systems. Overall, this manuscript was well written, the experiments carefully designed and performed, and the conclusions were supported by the data. This work should be of interest to the readership of PLOS ONE and I recommend to accept this manuscript for publication.

Reply: Thank you for working with our manuscript and the positive assessment.

Reviewer #2: Suggestions and questions:

1. Tetracycline is degraded by by ultraviolet radiation, and also (less efficiently) by visible light. As a consequence, reliable monitoring of tetracycline in the environment should ideally detect chemical forms derived from tetracycline. Such forms have usually lost antibiotic activity. To mimic this situation, the authors might test whether the biosensor described in this manuscript is able to detect autoclaved chlortetracycline, which has inducing activity but not antibiotic activity.

Reply: Thank you for working with our manuscript and the helpful comments. Indeed the tet –repressor is known to bind to chemically modified forms of Tc which can be pharmacologically relevant forms and degradation products. This information has now been provided in the introduction of the manuscript: “The TetR has been shown to bind efficiently to several chemically modified forms of tetracycline as well, including pharmacologically relevant forms and degradation products [32,37,38].”

2. Papers published in the 1980's (e. g., Moyed et al. 1983) showed that overexpression of tetracycline resistance proteins (e. g., from a multicopy plasmid) reduce tetracycline resistance. Should this phenomenon be taken into account? How important is plasmid copy number to make sure that the tetracycline sensor will work?

Reply: This is a very relevant comment. We have taken this up and expanded it even further also considering general genetic stability of biosensor bacteria. In the revised manuscript we added one sentence to the conclusion section proposing to integrate the sensor elements into the bacterial genome, instead of provided them on plasmids: “In order to increase genetic stability of the biosensor bacteria and avoid potential fluctuations in plasmid copy numbers under various growth condition, the sensor components could be integrated into the bacterial genome.”

Comments about writing (line numbers are tentative and have been identified by the reviewer): 

3. The introduction is very long. The description of the arabinose biosensor that provided the background for this study could be shortened without loss of clarity (especially considering that a figure describing the sensor is provided).

Reply: We have shortened the first paragraph of the introduction, which provides this information, by more than 15% and also shortened the tetracycline part.

4. Page 3, line 6. Because epigenetic biosensors are rare in bacteria, you might give some more details about, for instance indicating that the previously described sensors (refs. 9 and 12) are based on DNA methylation by two model DNA methyltransferases, CcrM and Dam.

Reply: This has been included as proposed.

5. Page 2, line 14. Remove "the" before "CcrM".

Reply: This has been corrected (on p. 3, this presumably was intended) and also at another similar occasion in the discussion section.

6. Page 7, first paragraph. Flow cytometry has been widely used in eukaryotic epigenetic sensors and also in the prokaryotic sensor described in ref. 12. You might mention this fact to further support your strategy in this study (and also to be fair to the literature).

Reply: The sentence has been rewritten to clarify that the aim was to use flow cytometry for our system. We do not claim invention of the application of flow cytometry and also not claim any priority. We hope that our intention is now appropriately transmitted. “This technology has been successfully used in the eukaryotic and prokaryotic sensors mentioned above and it was our first aim to establish the routine application of flow cytometry for the quantitative readout of our epigenetic memory systems, which previously was mainly based on fluorescence spectroscopy in whole-cell lysates [9].”

7. Page 11, Conclusions. Please revise the sentence that starts "This work...". The current sentence is difficult to read!

Reply: The sentence has been rewritten and it now reads: “In this work, we expanded an existing DNA methylation based arabinose sensor with memory function by designing a tetracycline sensor input system.”

---

## [Editor Report · Decision Letter 1]

21 Apr 2020

Development of an epigenetic tetracycline sensor system based on DNA methylation

PONE-D-20-06852R1

Dear Dr. Jeltsch,

We are pleased to inform you that your manuscript has been judged scientifically suitable for publication and will be formally accepted for publication once it complies with all outstanding technical requirements.

With kind regards,

Nathaniel A. Hathaway, Ph.D.

Academic Editor

PLOS ONE

---

## [Editor Report · Acceptance letter]

23 Apr 2020

PONE-D-20-06852R1 

Development of an epigenetic tetracycline sensor system based on DNA methylation 

Dear Dr. Jeltsch:

I am pleased to inform you that your manuscript has been deemed suitable for publication in PLOS ONE. Congratulations! Your manuscript is now with our production department. 

With kind regards,

on behalf of

Dr. Nathaniel A. Hathaway 

Academic Editor

PLOS ONE